# Application Studies for the Implementation of the Sustainability Charter in the Metropolitan City of Genoa

**Francesca Pirlone** [1,*] , **Ilenia Spadaro** [1,*] , **Cristiana Arzà** [2] , **Giovanna Lonati** [2] and **Piero Garibaldi** [2]

1   Department of Civil, Chemical and Environmental Engineering, University of Genoa, 16145 Genoa, Italy
2   Strategic Planning Office, Metropolitan City of Genoa, 16145 Genoa, Italy;
    cristiana.arza@cittametropolitana.genova.it (C.A.); giovanna.lonati@cittametropolitana.genova.it (G.L.);
    pianificazione.strategica@cittametropolitana.genova.it (P.G.)
*   Correspondence: francesca.pirlone@unige.it (F.P.); ilenia.spadaro@unige.it (I.S.);
    Tel.: +39-010-335-2820 (F.P.); +39-010-335-2820 (I.S.)

**Abstract:** Starting from Agenda 2030 and existing tools in the field of sustainability, this research defines the guidelines for a new Sustainability Charter created for a metropolitan-level city. These guidelines are then applied to the case study of the metropolitan city of Genoa. The paper reports, therefore, application studies for the implementation of the Sustainability Charter in the metropolitan city of Genoa. Funded by the Ministry of the Environment and the Protection of Territory and Sea, the Sustainability Charter of the Metropolitan City of Genoa, which we present here, is developed as part of "Agenda 2030, the Sustainable Metropolitan Agenda of the Metropolitan City of Genoa: moving towards sustainable metropolitan spaces". This research has led to the implementation of a concrete product the entire citizenship can benefit from. The new proposed tool is oriented towards the application of sustainability in urban planning and management in order to reduce environmental impacts and promote a proper and better quality of life: a driving force for sustainable urban development. Sustainability as a tool to safeguard the cultural and environmental heritage and the economic system, which can represent a new opportunity for the development of competitiveness, innovation and employment.

**Keywords:** Sustainability Charter; participation; incentives; urban planning; quality of life

## 1. Introduction

Global warming, the overexploitation of natural resources and the progressive ageing of the population are the challenges that humanity must face during future decades to prevent serious and irreparable damages to the Earth. The only turning point is to plan and manage our cities in a sustainable way.

Sustainability, as it is known, was introduced by Bruntland in 1987, asserting that sustainable development is a "Development that meets the need of the present without compromising the ability of future generations to satisfy their own" [1].

After 1987, a milestone was the introduction of Agenda 21, which came out from the Earth Summit in Rio de Janeiro, literally 'things to do in the 21st century', to translate the theoretical assumptions of sustainable development into actions [2–4]. Agenda 21 identified the instruments to combine environmental, economic and social development through the sharing of medium-long term scenarios and the assumption of responsibility by all social actors. "A mechanism, a path, a working method, a technical and cultural proposal that lays the foundations for stimulating local actions aimed at achieving and verifying the objectives of sustainable local development, concerted with the local community" [5–7].

Since 1992, Agenda 21 processes have been introduced at different national, regional, provincial/departmental and municipal levels, and thanks to these tools, different stakeholders have been gathered around a table. Since then, the population has started to play a fundamental role. Twenty years later, in 2012 in Rio de Janeiro, the situation of the

international level towards sustainability was analysed and monitored, but there was still a long way to go.

In 2015, at the Paris Climate Conference (COP21), many countries adopted the first universal and legally binding global climate change agreement. The agreement defines a global action plan. The first is the United Nations 2030 Agenda for Sustainable Development, the result of a complex process launched by the "Rio + 20" conference.

The 2030 Agenda for Sustainable Development, 'Transforming our World', based on the Sustainable Development Goals, expresses a clear judgment about the unsustainability of the current development model, not only on the environmental level, but also on the economic and social one. The implementation of the Agenda requires a massive participation of all components of the society. It is a process that involves citizens, companies and institutions in the definition of a new governance model that aims at sustainable development and puts innovation and technology at the service of society.

This tool pays attention to the role of cities, regions and local authorities. The agreement recognizes the role of stakeholders who are not part of the agreement in addressing climate change, such as cities, other subnational entities, civil society, the private sector and others, but are invited to intensify their efforts and support initiatives to reduce emissions, build resilience and reduce vulnerability to the negative effects of climate change and to maintain and promote regional and international cooperation. The Agenda considers 17 goals. Each goal has specific targets to be reached in the next 15 years. For example, goal 11, "Making cities and human settlements inclusive, safe, long-lasting and sustainable", includes a series of sub-objectives that are directly linked to a circular approach at the urban level, including: "increasing inclusive and sustainable urbanization" [8–10].

The aim of this challenge is to involve communities around the world to improve ever more the life of the planet and its inhabitants.

Planning a city and its priority issues in a sustainable way becomes the real challenge [11–14], not only for proper urban growth but also in relation to climate change [15,16].

The following year, 2016, the United Nations Habitat III conference in Quito drew up the new Urban Agenda, which contained guidelines for making cities around the world more inclusive, greener, safer and even more prosperous. It has been an essential element in the implementation of the 2030 Agenda for Sustainable Development and other reform agendas, in particular the Paris Agreement. The Urban Agenda presents three commitments focused on achieving the global objectives of the new urban agenda:

1.  Realize the new urban agenda (the action plans for the 12 priority themes are being developed);
2.  Develop a global and harmonized definition of cities (elaboration of an online database, a global list of cities and their main characteristics);
3.  Promote cooperation among the cities in the field of sustainable urban development (cities around the world are encouraged to establish a link with one or more partner cities to develop and implement action plans at local level and projects on common priorities) [17–19].

The concept of the circular economy [20–24], which is intended to create circular cities, where circularity becomes an element of achieving sustainability [8,25–27], is introduced in the Urban Agenda.

To make the concept of sustainability more concrete, other tools have been inserted, such as the Sustainability Charter, good practices, and indicators.

In USA, the Sustainability Charter has been in place for several years. It is a fluid and easy-to-read document. The Sustainability Charter is the result of research, expert advice and community input. It presents a vision of the community that meets the social, cultural, economic and environmental needs of actual residents, while ensuring those needs can continue to be met for future residents.

The Sustainability Charter should inspire all residents to think, make decisions and act with an eye to the future while living their daily lives. It is important for providing

guidelines for the city's decision making. It is a high-level policy to guide the organization and the community towards a sustainable future [28–30].

A good practice is "... an action, exportable to other realities, which allows a Municipality, a community or any local administration to move towards forms of sustainable management at the local level" [1].

Therefore, a practice is considered "good" if it responds to the idea of sustainability, considered as an essential factor of development capable of meeting "... the needs of the present, without compromising the ability of future generations to satisfy their own" [1].

Finally, indicators are important tools for the assessment of the relevance of actions, the evaluation of their potential results, the monitoring of their development and for their comparison with set objectives, paying attention to resources and therefore to the efficiency of the actions developed [31–35].

Starting from the existing tools [36–38], listed above, the paper reports research aimed at defining guidelines for a new Sustainability Charter, created at a metropolitan city level. These guidelines are then applied to the case study of the metropolitan city of Genoa. The Sustainability Charter of the Metropolitan City of Genoa (presented in the document) is part of the 2030 Agenda, "the Sustainable Metropolitan Agenda of the Metropolitan City of Genoa: towards sustainable metropolitan spaces", and it is financed by the Ministry of the Environment and the Protection of Territory and Sea. This application has led to the creation of a concrete product all citizens can benefit from. The proposed new tool is conceived thanks to the participation of the various stakeholders present in the metropolitan area, the proposal of incentives and the use of technology. The Sustainability Charter conceived in this way makes it possible to achieve sustainability in urban planning and management [39,40] and to improve the environmental, economic and social impacts of the activities that are usually carried out: mobility, catering, sports-related etc. Through the Charter, an improvement in the quality of daily life of inhabitants is therefore also achieved.

## 2. Methodology

### 2.1. Basic Aspects of the Sustainability Charter

This subsection describes the basic aspects of the structured approach (see Section 2.2) to achieve the goals of the new Sustainability Charter at the metropolitan level.

The metropolitan level constitutes the first fundamental point [41–43].

Metropolitan cities are the most important urban areas in the world and are directly involved in achieving the goals of sustainable development, the global transition to a circular economy and the different goals set by the 2030 Agenda [44–48].

The metropolitan cities are entrusted with strategic planning, the direction of economic and social development, the coordination of overall government action of their territory [49–53]. In Italy, a country where the guidelines are applied, on the occasion of the G7 Environment (Bologna, 2017), the Mayors of the Italian metropolitan cities signed the Bologna Charter for the Environment 'Metropolitan cities for sustainable development', which identifies eight environmental themes to work on a metropolitan scale: sustainable use of ground; circular economy; adaptation to climate change and risk reduction; energetic transition; air quality; water quality; ecosystems, urban greenery and biodiversity protection; and sustainable mobility. By signing the Bologna Charter for the Environment, the metropolitan cities have strongly committed to chase the principles and general objectives of the Charter by integrating them into the strategic visions and statutes of their cities, and adapting them to local contexts. In particular, they wanted to start the process of building a metropolitan agenda for sustainable development in each city [9]. For Italian metropolitan cities, but also for the other metropolitan cities of the world, this means opening up to the future with a new mentality, giving space to local leaders in order to face in-progress transformations with different perspectives based on sustainability.

The proposed new Sustainability Charter is part of the framework of sustainability, a priority aspect that emerges in the guidelines of the Strategic Plans of Metropolitan Cities.

Among the purposes indicated in the strategic plans, is the coordination of the lower levels and the engagement of the territory to create a sense of belonging to the metropolitan city.

Therefore, the new Sustainability Charter intends to stimulate citizens and municipalities to feel themselves as 'members' of a common project, to increase their sense of community and responsibility towards their city, to promote the diffusion of good practices and the formation of a virtuous system that aims at sustainability [54,55]. The concept of "partner" is important in this context, to feel an integral part of a community. Today agreements generally concern just the relationships between a single company providing the service and a customer. Thanks to the Metropolitan City Charter, the services presented in a metropolitan area are put online with an immediate access for the citizens.

Essential points for the Charter are:

- Promoting transparency;
- Declaration of formal commitment of managers to users;
- Identifying the fundamental principles to be followed by the operator in managing the Integrated Water Service;
- Identifying service quality standards;
- Defining the relationship between operator and users with regard to participation and information rights and establishing user complaint procedures.

To increase the sense of community and responsibility, it is essential to consider the concept of involvement and participation in the editing of the new Sustainability Charter. The logic of the Quintuple Helix and the new 5.0 society can allow for collaboration between innovation, technology and society in order to improve the quality of life of urban areas, enhance production, lead to environmental, landscape and cultural excellence, and thus contribute also to the mitigation of the phenomenon of depopulation of inland areas of the metropolitan area.

The concept of a Quintuple Helix represents an evolution of what Etzkovitz-Leydersdorff theorized in the 90s. Starting therefore from the Triple Helix [56,57], where the main actors—capable of creating a favourable context for the transfer of knowledge—were universities, private sector and public administration, today, the Quadruple [58,59] or Quintuple Helix is applied. There are five key players in creating a distributed and collective intelligence, which is fundamental for the sustainable development of a city and its territory: Public Administration, Research, Companies, Population and Associations or Investors (Figure 1).

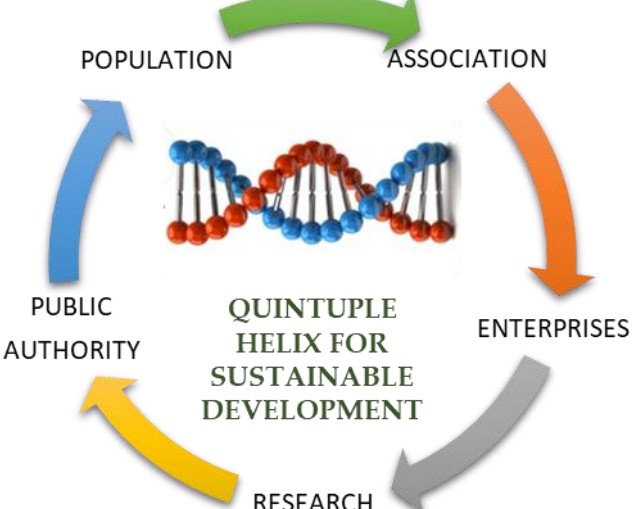

**Figure 1.** The concept of the innovation helix framework.

The idea of a 5.0 Company, born in Japan, captures these transformations by placing people at the centre of the innovation process. In a 5.0 Company, many processes will start

from the needs of the final customer: we will pay for what we actually consume. This is a step that helps us to understand how digital technology is changing the future of business in a sustainable way and how urgent it is to adapt our business models [60–62].

The last tools identified are incentives [63,64].

In China, for example, the Social Credit System is meant to provide an answer to the problem of lack of trust in the market. Proponents argue that it will help eliminate problems such as food safety issues, intellectual property theft, violation of labor law, financial infidelity, and counterfeit goods. It is an example of China's "top-level design" approach. If the Social Credit System is implemented as envisioned, it will constitute a new way of controlling both the behavior of individuals and of businesses. Once implemented, the system will manage rewards and punishments for businesses, institutions and individuals on the basis of their economic and personal behavior. Punishments for poor social credit include increased audits and government inspections for businesses, reduced employment prospects, travel bans, exclusion from private schools, slow internet connection, exclusion from high-prestige work, exclusion from hotels, and public shaming. Rewards for positive social credit include less frequent inspections and audits for businesses, fast-tracked approvals for government services, discounts on energy bills, being able to rent bikes and hotels without payment of a deposit, better interest rates at banks, and tax breaks [65].

However, in the Sustainability Charter, the incentives are promoted as a positive lever to promote sustainability. The incentive is not economic but consists of a sustainable prize, a service offered in the metropolitan area itself (Figure 2). The metropolitan member, as indicated here above, can participate in the Sustainability Charter both by using sustainable services the incentives are associated to, or as the proposer of a sustainable service/product that constitutes an incentive for others taking part in the project [66,67].

## CHARTER OF SUSTAINABLE SERVICES

Once the customer has reached the minimum number of points / stamps to receive the incentive, he can choose the one he prefers from the rewards catalog. Based on the number of services and best practices implemented, the size / number / value of the prize received will change.

### Discounts

The holders of the Service Card can obtain discounts on many products and services offered

### Incentives

By the Service Card it is possible to participate in the collection of points and choose the prizes from the catalog.

### Events

By the Service Charter, it is possible to participate in themed meetings and insights on sustainability and the circular economy to receive personalized advice.

### Welfare

By the Service Charter it is possible to discover a world of services and products for your welfare.

**Figure 2.** Sustainability Charter and the reward system.

Incentives, loyalty, points collection, promotions and discounts represent the most innovative solutions to motivate and push populations towards good behaviours and useful actions to achieve sustainable development.

These solutions are clearly in contrast with "restrictive" strategies that often act by imposing constraints. Incentives, on the other hand, by definition, are positive reinforcements, useful for motivating action, becoming an opportunity and a lever for sustainability [68–70]. The promotion of discounts on LPT tickets, on waste tax or on city services as a result of virtuous behaviour can contribute, together with a mix of restrictive policies, to sustainable development. Thinking about the mobility sector, for instance, an incentive is

effective if it motivates individual travellers to change their travel behaviour in order to achieve the objectives identified by local administrations: reduce use of private cars, protect the environment, promote the comfort of communities using soft mobility during urban travel [71,72]. Currently, for example, in mobility policies we find more disincentives rather than incentives. Restrictive or sanctioning actions are often implemented, and increased, to discourage the use of private vehicles in favour of public ones (ZTL, pedestrianization, parking areas with high costs, fines, etc.), and traffic calming measures are a planned way to reduce driving speed and improve street safety and quality of life in urban areas (narrow lanes, bumps, roundabouts, zone 30, etc.). What has been described for mobility is also valid for other priority sectors such as energy, waste, tourism etc.

Moreover, this research intends to define and create also a physical support, that is a "card" for each citizen adhering to the Sustainability Charter, which, through the monitoring of behavior and the acquisition of points, rewards virtuous behaviors, allowing also citizens to understand the sustainable services offered in the metropolitan area. The system also takes its inspiration from the theory of "nudge", a gentle push, which leads people to take better decisions without being obliged, but simply by changing the way they make their choices, by acting, for example, through a sort of social pressure [73,74].

In order to allow the sustainability of the Sustainability Charter, and therefore for the achievement of the goals it intends to reach, the participation of all public and private actors who live or transit in the metropolitan area is fundamental. Thanks to the involvement and awareness of citizens in the long term, there is a ripple effect that attracts more and more interest, and therefore generates a circuit of virtuous behaviours, both for the provision of sustainable services (sustainable mobility, separate waste collection, domestic composting, courses on sustainability, sales of products at Km0 etc.), and for related incentives (better prizes that can be obtained using sustainable services) [75,76].

### 2.2. Guidelines for a Genoa Metropolitan Sustainability Charter

This subsection reports the methodological approach developed in the research that conduces to the drafting of operational guidelines for structuring a Sustainability Charter to be developed at a metropolitan level. The guidelines introduce procedures and useful tools that other metropolitan administrations could use to create their own Sustainability Charter.

The approach was structured to be scalable and adaptable to other administrative realities. Thanks to the definition of incentive policies, it intends to involve municipalities and the population through the adoption of sustainable behaviours and purchases [77,78]. Indeed, the Sustainability Charter relates those who provide services in the metropolitan area (proposing actors) and the 'Metropolitan Member', i.e., those who benefit from them (individuals to whom the action is aimed), with the introduction of a points collection system (rules and incentives) that rewards good practices, and therefore consumption and virtuous behaviours [79,80].

Metropolitan Members can be involved in the Sustainability Charter either individually (as resident, student, worker, or tourist), or collectively as a member of a community (Municipality, School/University, sportive, religious, non-profit association, etc.) The Metropolitan Member, in aggregate form, can receive and use sustainable services but, in turn, can be a supplier of sustainable services and incentives for its members. The "Aggregate Metropolitan Member" can be an interesting stakeholder in order to improve awareness, by creating a reference point for the community and therefore a place for exchange, and the spread of good practices and incentives (Figure 3).

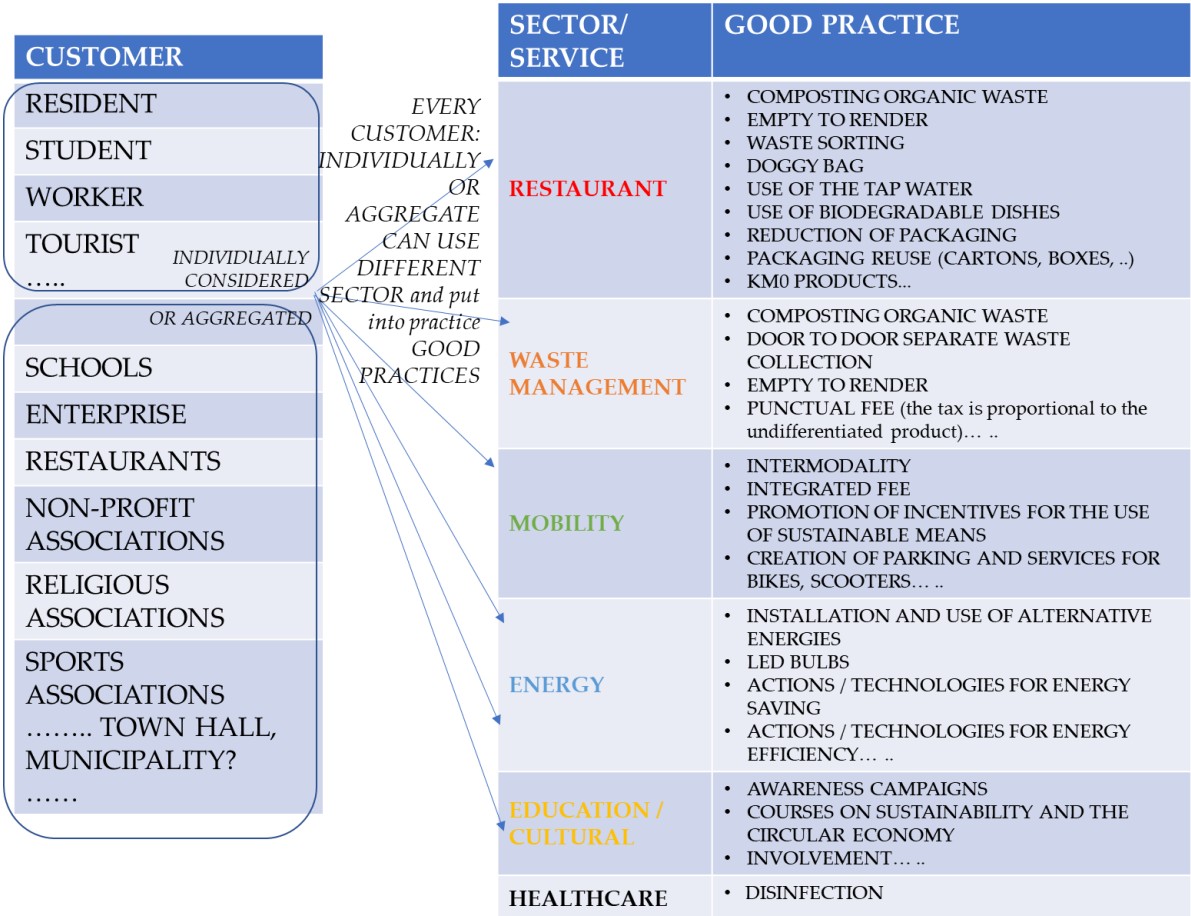

**Figure 3.** Relationships between customers, sector/service and good practices.

The research is finalized by defining and implementing a Sustainability Charter, structured in eight different phases (see Figure 4).

Phase 1: definition of the strategy and involvement of a territory stakeholder: according to the objectives of the Metropolitan Agenda for Sustainable Development, concerns definition of the strategy to identify and involve providers of sustainable sectors of incentives, through a participatory dialogue on the one hand, with the main stakeholders involved (municipalities, public and private companies, private citizens, etc.), and on the other, with those who can support the operational phases of implementing the service charter (municipalities, post offices, etc.)

Phase 2: definition and choice of essential sectors/services: is focused on the choice of essential sector or services (waste management, mobility, energy, education/cultural, catering, sport, tourism, etc.) to be able to unite many good practices and, therefore, virtuous behaviours of the Metropolitan Member. It must be emphasized that the incentives do not depend and are not linked to particular services, but, on the contrary, can involve different ones. For example, a member who joins the Sustainability Charter and conducts home composting can receive, as an incentive, a free visit to a museum and a discount on the local public transport ticket to reach it; a school who promotes training courses on sustainability can receive composters, and gift students vouchers to visit a museum; a restaurant that uses the returnable empty service can, for example, receive discounted jugs or doggy bags for its customers; a company that creates bicycle parking spaces can obtain a discount for the use of bike sharing, for the purchase of electric bikes to be distributed among its employees, or a discount for visiting or attending particular events; or a tourist who buys a public transportation pass can receive a discount for the purchase of products at km0.

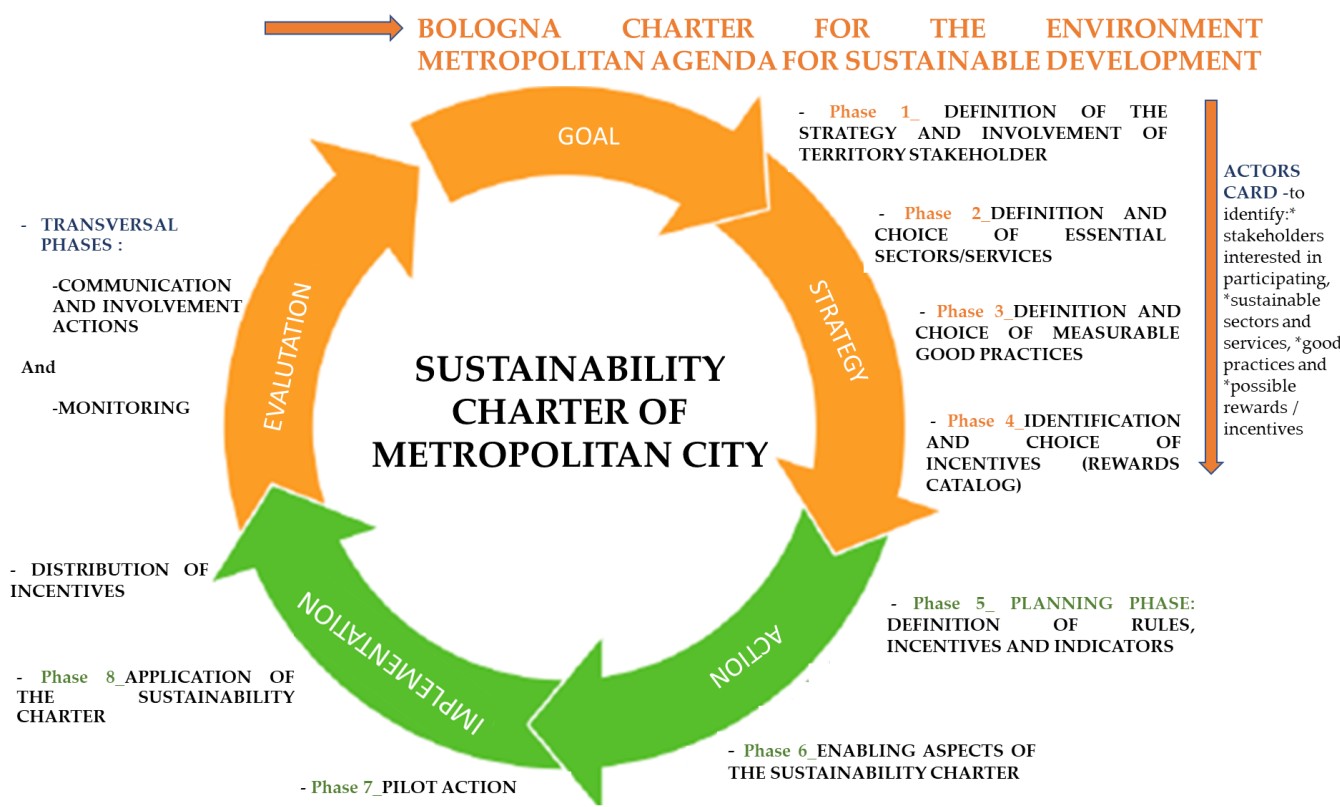

**Figure 4.** Phases for the creation of a Sustainability Charter.

Phase 3: definition and choice of measurable good practices: considers the choice of good practices. Within the sectors identified as essential, it is necessary to select good practices that are considered a priority in achieving higher levels of sustainability. An important aspect to be considered in the selection of such good practices is that they need to be measurable in order to be properly monitored. This is useful for associating a score to the virtuous behaviours and, thanks to the rules, creating the incentive system and thus the catalogue of rewards (Phase 4).

Phase 4: identification and choice of incentives: reports the rewards catalogue and the incentive system. The "Metropolitan Members" who adhere to the Sustainability Charter and participate in implementing virtuous behaviours can receive the sustainable incentives offered by the sustainable subjects that have joined the initiative, through a declaration of interest, "Actors' Card", shown below. These are discounts for the purchase of products or participation at events, visits to cultural sites for well-being, priority access to citizen services, or mobility vouchers (discounts on tickets and TPL passes, discounts on intermodality public transport, discounts on subscriptions of car and bike sharing rentals). In terms of approach, the minimum requirement to obtain a prize is to carry out two good practices in at least two different types of sector (mobility, energy, waste, etc.)

To allow Phases 2, 3 and 4 to be operational, an "Actors' card—recognition of good practices and incentives/rewards" was prepared in Excel format to return and systematize the answers. Thanks to this form, it is possible to identify the stakeholders (who supply sustainable services in the metropolitan area) interested in joining the program, the sustainable services, best practices and possible rewards/incentives that are available to those participating in the initiative.

The form is structured as follows: an introduction that illustrates what the Sustainability Charter is; a section dedicated to the data of the structure (company/organization); and a final section dedicated to good practices and any prizes that each participating structure wants to offer. In order to facilitate understanding of the participation mechanism,

an explanatory facsimile on measurable sustainable practices, i.e., actions which it was possible to associate points to, is attached to the questionnaire (Figure 5).

Phase 5: planning phase: concerns the planning aspect of defining the rules [81–84]. Thinking about the main sectors, we propose a set of rules the virtuous behaviours of the Metropolitan Member are associated to (implementation of good practices) with points that correspond to incentives/prizes (catalogue of prizes).

Under the coordination and supervison of the Metropolitan City of Genoa, which proceeds with involvement of the various stakeholders, for each main sector, the following aspects are defined (Figure 6):

- Goal—expected results, actions/best practices;
- Target (actors/proponents, who promote the action and people to whom the action is addressed);
- Timing;
- Incentives;
- Monitoring indicators.

Phase 6: enabling aspects of the Sustainability Charter: considers the enabling aspects of the Sustainability Charter (technology or other). Initially, in order to digitize the Sustainability Charter, various options are possible to analyze: from the creation of a totally new, special app to the recognition of existing apps that can be adapted to the logic of the identified methodological approach. It is, therefore, possible to conduct specific research on existing apps on the market, in order not to create additional tools, but to enhance and capitalize already active experiences and adapt them to the development of the Sustainability Charter. It is also possible to enhance IT platforms, perhaps already in the possession of the metropolitan administration, which can make visible the participants in the Sustainability Charter that provide sustainable services in the area. Information about the regulation, the points attributed to virtuous behavior and the catalog of rewards/incentives will then be reported on the platform.

| ACTORS CARD RECOGNITION GOOD PRACTICES AND INCENTIVES / REWARDS | |
|---|---|
| **DATA OF YOUR STRUCTURE** | |
| **Name** | |
| **Email** | |
| **Phone** | |
| **1_ Are you available to collaborate in the implementation of the Sustainability Charter?** *(Sign more than one choice)* | *NOTE: It is possible to participate as a service provider but also as a support to the initiative. In the first case, you can offer your customer service; in the second case you can allocate prizes, discounts, concessions to be made available for those who behave in a virtuous way in the context of the Sustainability Charter* |
| AS A SUSTAINABLE SERVICE PROVIDER? | |
| AS A SUPPORT TO THE IMPLEMENTATION OF THE SERVICE CHARTER? | |

| 2_ What sustainable sectors/service (s) do you offer in the Metropolitan area? | *NOTE: tick with a x* |
|---|---|
| WASTE MANAGEMENT | |
| MOBILITY | |
| ENERGY | |
| EDUCATION/CULTURAL | |
| RESTAURANT | |
| SPORTS | |
| TUORISTIC | |
| RELATED TO AGRICULTURE AND BREEDING | |
| Other (specify) | |
| | |
| **3_ Do you believe that your service meets the requirements of sustainability and why?** | *NOTE: brief explanation of how the service respects sustainability* |
| ENVIRONMENTAL | |
| SOCIAL | |
| ECONOMIC | |

**Figure 5.** *Cont.*

| 4_ What measurable actions / good practices can you offer in the Metropolitan area? | | | |
|---|---|---|---|
| - Measurable good practices<br><br>-Modality of quantification<br>*a……*<br>…<br>*b…..*<br>…<br>*c….*<br>…<br>*d….*<br>.. | *NOTE: look below examples of possible good practices and method of quantification* | | |
| | **Sector/ Service** | **Example of measurable good practices** | **Example of Method of quantification** |
| | Waste management | Returnable | 1 point for a number of bottles returned |
| | | Waste sorting | Sacks of undifferentiated (less sacks, more virtuous …) |
| | Restaurant | Use of doggy bags | 1 point for a doggy bag |
| | | Purchase of bulk products with their own containers | 1 point for a total of loose purchases |
| | Mobility | Use of bike sharing | 1 point for each bike rental |
| | | Purchase of public transport passes | Points according to the duration of the subscription |
| | Cultural | Participation in events related to the theme of sustainability | 1 point for each event |
| | | Participation in beach cleaning initiatives, parks, .. | Tot points for each initiative |
| | Commercial | Use of water dispensers with their own containers | 1 point for every number of container fills |
| | | Purchase / donation of second-hand products | 1 point for each amount of expenditure |
| | | Purchase of organic, eco-friendly | 1 point for each amount of expenditure |

| 5_ What rewards / incentives do you intend to offer to participants in the Sustainability Charter? | | *NOTE: Example of possible rewards / incentives: if you are a structure that operates in sustainable mobility you can offer an incentive / reward concerning your service such as eg. discounted daily or monthly ticket, or a reward that you have available thanks to your existing agreement, eg. discount for ticket to the theater, …)* |
|---|---|---|
| **Rewards / incentives offered** | **Value** | *NOTE: Please indicate on the left any prize and its value according to the bands below:* |
| | | **First bracket: € 0-25** |
| *a.* | | **Second bracket: € 25-50** |
| *b.* | | **Third bracket: € 50-100** |

**Figure 5.** Extract of an Actors' card—recognition of good practices and incentives/awards.

| GOAL | Actions/ GOOD PRACTICE | TARGET | | RULE | INCENTIVES | INDICATORS |
|---|---|---|---|---|---|---|
| | | **Proposing Actors (SERVICE PROVIDERS)** | **Acord to Whom the Action Is Addressed (CUSTOMERS)** | | | |
| Sector 1<br>… | | | | | | |

**Figure 6.** Phase 5: planning.

Phase 7: pilot action: regards testing. Through specific pilot actions in the area under study, the effectiveness of the implemented system is checked.

Phase 8: application of the Sustainability Charter: consists in the implementation of the approach, i.e., the implementation of the Sustainability Charter. In this phase, the proposed model is applied to the entire metropolitan area.

Two other transversal phases were then defined for the whole process: communication and involvement actions and monitoring.

The first is addressed to the inhabitants to raise their awareness, train and educate them about the importance of sustainable development and to illustrate to them the purpose and

instructions for using the Sustainability Charter. Furthermore, this Charter also addresses possible service providers to encourage them to promote increasingly sustainable actions.

In the monitoring phase, specific indicators are defined in order to objectively evaluate the various actions proposed, before and after the implementation of the Sustainability Charter. The continuous monitoring of the activities, starting from the testing phase, is essential to be able to recalibrate the system in case of failure to achieve the set objectives.

In the project, monitoring has two levels of application. The first level is used by the Metropolitan City to measure progress towards the achievement of the goals of the Metropolitan Agenda for Sustainable Development and the Sustainability Charter. They are distinguished for the impact areas (Figure 7) identified by the Bologna Charter for the Environment's 'Metropolitan Agenda for sustainable development' (see Section 2.1).

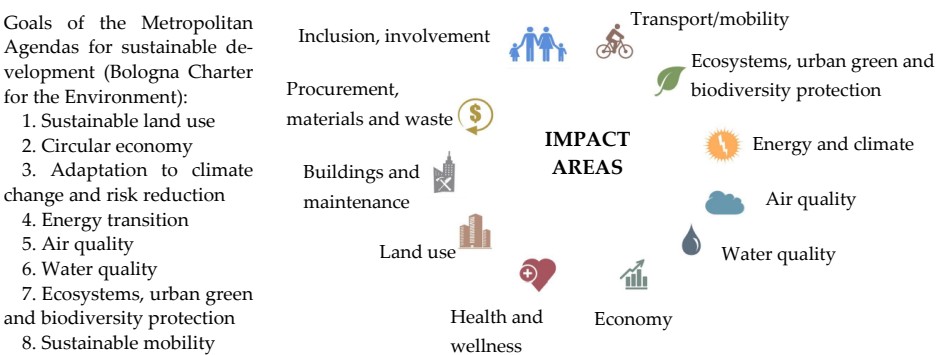

Goals of the Metropolitan Agendas for sustainable development (Bologna Charter for the Environment):
1. Sustainable land use
2. Circular economy
3. Adaptation to climate change and risk reduction
4. Energy transition
5. Air quality
6. Water quality
7. Ecosystems, urban green and biodiversity protection
8. Sustainable mobility

**Figure 7.** Goals and impact areas identified by the Bologna Charter for the Environment.

In order to monitor the achievement of the sustainability goals for each implemented action/good practice, it is possible to consider, for example, their status of application with respect to the various impact areas investigated (see diagram shown in Figure 8). This approach, related to monitoring, therefore provides for the use of a traffic light system (for a qualitative evaluation) accompanied by information related to the state of implementation (not yet applied during project phase, started in plan).

**Figure 8.** Approach for monitoring the objectives of the Metropolitan Agenda for Sustainable Development achieved thanks to the Sustainability Charter.

The second level of application for monitoring consists in assessing whether the Metropolitan Member has implemented the good practice, in order to award the points associated with a sustainable behaviour.

## 3. Application and Results: Implementation of the Sustainability Charter in the Metropolitan City of Genoa

This section implements the guidelines illustrated in Section 2 to the case study of the Metropolitan City of Genoa. In particular, the application is reported with the various experiments that led to the creation of the Genoa Sustainability Charter. This product became operational in September 2021. The Metropolitan City of Genoa (Liguria Region, in the northwest of Italy) is one of Italy's fourteen metropolitan cities; it is a large area territorially (surface 1833.79 km$^2$, population 823,612, as of 2021, with a population density of 465.76 inhabitants per km$^2$) which has been operational since 1 January 2015, with the same territory of the province, made up of 67 municipalities. Genoa was chosen because it has defined a participatory strategy in the implementation of its political instruments, involving local stakeholders (associations, public-private partnerships, research institutes, companies, etc.) In addition, the Metropolitan City of Genoa has created a website called Fuori Genova (Outside Genoa) that also offers the possibility of discovering development opportunities by creating new networks. "The site is above all a database which collects different information about the whole metropolitan area, organized according to the following categories: public spaces, companies, artisan, civil associations, historical sites, parks, natural sites and tourism. It is a contact point between public and private actors, where it is possible to share personal opinions about metropolitan projects and policies. All information are geo-referenced on an interactive map" [85].

Below is reported the realization of the Sustainability Charter of the Metropolitan City of Genoa, which, as already illustrated in the section dedicated to the approach, includes eight consecutive phases and two transversal ones for the whole process.

First of all, for the realization of the Sustainability Charter of the Metropolitan City of Genoa, the involvement of the territory (and therefore of the various sustainable services/structures present) was necessary. Thanks to the administration of the "Actors' Card" (Figure 5), the various manifestations of interest in participating in the initiative—as providers of sustainable services, rewarding, or both—were collected (Phase 1 and transversal phase: communication and involvement actions). Through this form, information was also collected on existing good practices, distinct to the chosen essential services, and on the incentives/rewards promoted by the Sustainability Charter (Phase 2).

Phases 1 and 2 are concentrated on the involvement of actors (public and private companies, associations, museums, etc.) present in the metropolitan-political area, who were interested in taking part in the project. Through a participatory dialogue, the main services were selected, which, for the first application of the Sustainability Charter, are: mobility, energy, waste management, education/culture and catering/food. For which, the good practices to be awarded (Phase 3) and the incentives (Phase 4) were defined. During the third phase, the monitorable good practices considered to be a priority for achieving higher levels of sustainability were selected. It was then possible to assign a score to these good practices and, thanks to the rules (Phase 5), define the prizes through an incentive system (Phase 4).

In Phase 6, the Sustainability Charter of the Metropolitan City of Genoa oversaw an initial experiment through the creation of a points collection card, in a paper format to be given to each virtuous citizen. The paper mode was initially considered easier to use, especially by elderly people or people not used to technological applications.

This first application was implemented in October 2020, during the European Mobility Week 2020 "Zero emissions, mobility for all". The Sustainability Charter was therefore presented for the first time as part of an initiative organized within the Sustainable Development Festival. It is a non-competitive, participatory pedal race, an event open to all citizens that takes place on a city path, of about 12 km, between Chiavari and Lavagna.

During the Festival, participants were then trained and made aware of the Sustainability Charter project. Subsequently, great attention was paid to the mobility sector and how to improve it in order to increase the quality of urban life and the comfortable living of inhabitants. Stimulating the interest of citizens, involving them after the event in environmental issues and learning about their habits in daily travel was important for promoting virtuous behaviour and identifying possible, interested stakeholders. Upon registration, all participants were also given the illustrative leaflet of the Sustainability Charter and the prototype of the card for collecting points (Figure 9).

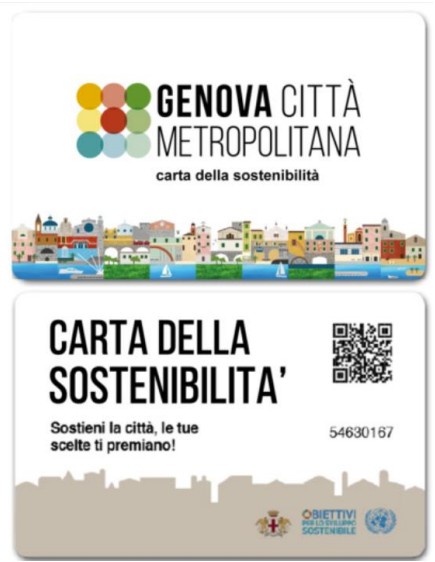

| SERVIZI | BUONE PRATICHE | |
|---|---|---|
| GESTIONE DEI RIFIUTI | | |
| MOBILITA' | | |
| ENERGIA | | |
| ISTRUZIONE/ CULTURALI | | |
| RISTORAZIONE | | |
| SPORTIVI | | |
| TURISTICI | | |
| LEGATI ALL'AGRICOLTURA ALL'ALLEVAMENTO | | |

SECTORS/SERVICES/GOOD PRACTICES

-WASTE MANAGEMENT

-MOBILITY

-ENERGY

-EDUCATION/CULTURAL

-RESTAURANT

-SPORTS

-TOURISTIC

RELATED TO AGRICULTURE AND BREEDING

**Figure 9.** The Sustainability Charter (front and back): October 2020 first experimentation.

There was also a subsequent experimental phase aimed at involving some CEAs, "Environmental Education Centres'", which promoted the Sustainability Charter in the territory of the Metropolitan City of Genoa through various actions.

Therefore, in phase 6, the Metropolitan City of Genoa wanted to promote a Sustainability Charter as a digital incentive and reward tool for every citizen for environmentally friendly behaviour using a reward points system [86–88]. The goal of the project was to spread awareness and promote social and entrepreneurial activity on sustainability issues, as well as through the broader involvement of citizens and civil society.

Initially, a survey was carried out in research focused on existing apps on the market.

The app developed in the PRINCE project, "PRemicity and INCEntives for modal change", was considered. PRINCE was funded by the national experimental program of sustainable home-school and home-work mobility, promoted by the Ministry of the Environment and the Protection of the Territory and the Sea (2018, ongoing). The project leader is the Municipality of Genoa—Direction mobility and the University of Genoa is a partner. This project gives incentives to students who travel in sustainable transportation to university centres [89,90]. The app created was, however, too specific to sustainable mobility and difficult to be adapted to the purposes of the Charter.

The EcoAttivi app was subsequently considered. EcoAttivi is an app that already involves several essential sectors chosen by the Sustainability Charter, in which the community of citizens is committed to increasing their environmental knowledge, moving in a sustainable way, recycling differentiated materials in the correct way and to ecological islands, reducing their waste with household composting or by purchasing less packaging, as well as frequenting libraries. The sharing of values is the main lever used to support the change of established habits and give meaning and purpose to new lifestyles. Therefore, this app has both the mission of the Sustainability Charter, aimed at educating the citizens

of the metropolitan city to virtuous behaviour, and also the methodological approach of the Charter itself, which is based on participation and on an incentive/reward system.

The Sustainability Charter is therefore developed by capitalizing on those potentially already developed in the app EcoAttivi, though also introducing some reinterpretations and adaptations.

The sustainable practices that allow for accumulation of points have been integrated into the existing EcoAttivi app.

The good practices related to the mobility sector are:

- Monitoring of km travelled on foot, or by bicycle, based on speed detection (mobility function);
- Use of public transport: scanning of QR codes placed on public transport (QR code function).

The good practices related to the waste management sector are:

- Home composting: sending a photo of your composter (photo sending function);
- Use of the water house (photo sending function).

The good deeds ascribable to the energy sector are:

- Production of energy from alternative sources: sending a photo of one's own renewable energy plant (photo sending function).

The good actions attributable to the education/cultural sector are:

- Borrowing books from a library: scan QR code on bookmarks provided by participating libraries (QR code function);
- Attendance of places of interest as an opportunity to raise awareness of issues related to theme of sustainability: QR code scanning with ecostop, as well as events on issues related to sustainability available at the Genoa aquarium and in three selected ecological islands;
- Participation at events: scanning QR codes indicated during events on the topic of sustainability, such as those of the National Sustainability Festival (14 October 2021) or conferences on the topic (QR Code function);
- Participation in quizzes: questionnaires consisting of 10 questions each, referring to the knowledge of sustainable actions and especially to the existence of good practices in the Genoese metropolitan area. This has the scope of raising awareness in the citizen to adopt virtuous behaviours in the different sectors of urban areas (mobility, waste, energy, etc.) and providing citizens knowledge of the presence of good practices in the area, while also being very easy to use (quiz function). The quizzes that have been developed for the metropolitan area of Genoa refer to all the main sectors investigated.

The good actions related to the restaurant/food sector, in addition to the use of the water box, require, for example, the sending of photos proving the use of the doggy bag and participating in the specific quiz.

Using the existing app, it was possible to include actions already active at the national level that allow obtaining points related to the involvement and spread of the project, which are:

- Inviting a friend: share your code to invite a new user (invite a friend function);
- Missions: completing sustainable missions, for example, obtaining a few points in a certain amount of time or carrying out a specific action (missions function).

Phase 4, "Rewards Catalogue and Incentive System", and Phase 5, "Rules Definition Planning", were developed in parallel. Figure 10 shows good practices, the associated scores and the method of accruing the points used in the app.

| SECTOR/ SERVICE | GOOD PRACTICES/ TYPE OF ACTION | SCORE IN ECOPOINT | FREQUENCY | METHOD OF ECOPOINT ACCREDITATION |
|---|---|---|---|---|
| **MOBILITY** | Movements on foot or by bicycle | 10 | per km | ECOATTIVI App mobility function |
| | SEND PHOTOS (charging electric cars or bikes / car-sharing) | 600 | when the photo is sent | ECOATTIVI App SEND PHOTO function |
| | QUIZ | 5 | each correct answer (10 questions every Thursday) | ECOATTIVI App QUIZ function |
| | Use of public transport | 100 | for each trip (maximum one per day) | ECOATTIVI App QRCODE function (STICKERS ON BUS) |
| **WASTE MANAGEMENT** | SEND PHOTOS (separate collection or waste delivery to the ecological island) | 600 | when the photo is sent | ECOATTIVI App SEND PHOTO function |
| | Home composting | 600 | when the photo is sent | ECOATTIVI App SEND PHOTO function |
| | QUIZ | 5 | each correct answer (10 questions every Thursday) | ECOATTIVI App QUIZ function |
| **ENERGY** | SEND PHOTOS (solar panels) | 600 | when the photo is sent | ECOATTIVI App SEND PHOTO function |
| | QUIZ | 5 | each correct answer (10 questions every Thursday) | ECOATTIVI App QUIZ function |
| **EDUCATION /CULTURALS** | Loan books in the library | 100 | for each loan (maximum one per day) | ECOATTIVI App QRCODE function (BOOKMARK) |
| | Frequenting places of interest towar | 200 | at each visit (maximum once per user) | ECOATTIVI App QRCODE function |
| | Participation in events | 200 | for each event (maximum once a day) | ECOATTIVI App QRCODE function |
| | Invite a friend | 100 | per ogni persona a cui si invia l'invito a scaricare l'app | ECOATTIVI App INVITE A FRIEND function |
| **RESTAURANT/ FOOD (no food waste and water saving)** | Use of the water house | 10 | maximum once per user | ECOATTIVI App QRCODE function |
| | SEND PHOTOS (doggy bag) | 600 | when the photo is sent | ECOATTIVI App SEND PHOTO function |
| | QUIZ | 5 | each correct answer (10 questions every Thursday) | ECOATTIVI App QUIZ function |

**Figure 10.** Sustainability Charter of the Metropolitan City of Genoa: Implementation of Planning Phase.

Metropolitan Members who adhere to the Sustainability Charter, through the free EcoAttivi smartphone app, can thus accumulate points for each action (sustainable mobility, local public transport, book lending in the library, participation in sustainability events, quizzes and proposed challenges on the app), obtain virtual tickets and participate in the draw of prizes offered by the EcoAttivi competition (at national level where the prize is an electric car, which already exists) and in the local competition (newly introduced), where prizes are offered by the network of local stakeholders (Fiab, Val d'Aveto Dairy, Genoa Aquarium Foundation), surveyed thanks to the Actors Card.

Using the new app, two lottery tickets are issued every 100 points accumulated: a ticket for a local competition and a ticket for the national competition. Compared to the initial idea, not all participants are rewarded, but the value of the prizes is higher. The EcoAttivi app therefore quickly enabled an initial application of the initiative.

In September 2021, the Digital Sustainability Charter was launched using the EcoAttivi app and the related competition (Figure 11).

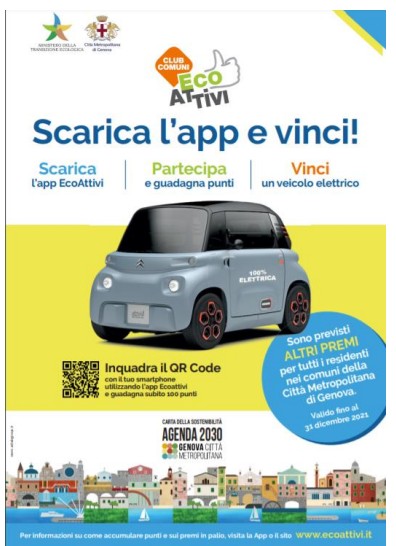
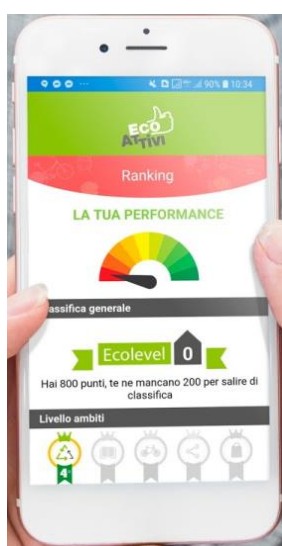

**Figure 11.** Launch of the digital sustainability Chart.

## 4. Discussion and Conclusions

The action of communication and involvement of stakeholders (population, public and private actors) within the metropolitan area constitutes the fundamental element of the new participatory tool of sustainability.

As expected in the approach, and applied in the implementation phase of the Genoa Sustainability Charter, local actors must be involved, made aware, educated about the importance of sustainable development and also trained on the aims and operational instructions for joining the project.

All the activities to define and implement the Metropolitan City of Genoa Charter started from the involvement of citizens and adapted to the results in terms of participation achieved.

The Sustainability Charter was, in fact, built around the actors involved, the specificities and local excellences present, and the prizes themselves represent a resource and source of pride in order to attract the population to participate and achieve sustainability, changing together harmful habits. Through this project, sustainable entities have one more opportunity to be more competitive and publicized.

Communication is, therefore, the tool that needs to be used to to involve the territory. During the creation of the Charter, specific documents, brochures and internet pages were provided, as well as different Events organized (for example during the National Sustainability Festival, Genoa Smart week etc.) where the objectives and operations of the Charter have been illustrated (Figure 12).

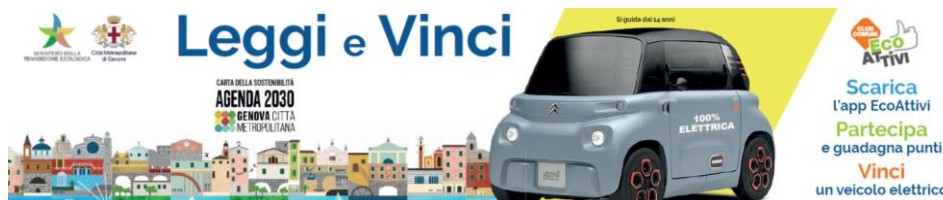

**Figure 12.** Information material prepared (by ACHAB S.r.l).

Another fundamental transversal action in the project is monitoring, which accompanied the creation of the Charter from the first months of defining the approach and during its implementation. This was conducted in order to objectively estimate the various actions before and after the implementation of the Sustainability Charter of the Metropolitan City of Genoa. Two levels of monitoring application have been applied.

The first level evaluated the achievement of the objectives of the Metropolitan Agenda for Sustainable Development and the Sustainability Charter, divided, as already described, for the impact areas identified by the Bologna Charter for the Environment, 'Metropolitan Agenda for sustainable development' [9]. As specified in Figure 13, for each good practice proposed and monitored, thanks to the use of the app, potential progress is assessed with respect to the areas of impact to be investigated.

| SECTOR SERVICE | GOOD PRACTICES/ ACTIONS | IMPACT AREAS | | | | | | |
|---|---|---|---|---|---|---|---|---|
| | | Inclusion/involvement | Mobility | Ecosystems | Energy/climate | Air and water quality | Circolar economy | Land use |
| MOBILITY | Movements on foot or by bicycle | 🙂 | 🙂 | 🙂 | 🙂 | 🙂 | 🙂 | 🙂 |
| | SEND PHOTOS (charging electric / car-sharing) | 🙂 | 🙂 | 🙂 | 🙂 | 🙂 | 🙂 | 🙂 |
| | QUIZ | 🙂 | 🙂 | 🙂 | 🙂 | 🙂 | 🙂 | 🙂 |
| | Use of public transport | 🙂 | 🙂 | 🙂 | 🙂 | 🙂 | 🙂 | 🙂 |
| WASTE MANAGEMENT | SEND PHOTOS (separate collection or ecological island) | 🙂 | ☹️ | 🙂 | 🙂 | 🙂 | 🙂 | 🙂 |
| | Home composting | 🙂 | ☹️ | 🙂 | 🙂 | 🙂 | 🙂 | 🙂 |
| | QUIZ | 🙂 | ☹️ | 🙂 | 🙂 | 🙂 | 🙂 | 🙂 |
| ENERGY | SEND PHOTOS (solar panels) | 🙂 | ☹️ | 🙂 | 🙂 | 🙂 | 🙂 | 🙂 |
| | QUIZ | 🙂 | 🙂 | 🙂 | 🙂 | 🙂 | 🙂 | 🙂 |
| EDUCATION /CULTURALS | Loan books in the library | 🙂 | ☹️ | 🙂 | 🙂 | 🙂 | 🙂 | 🙂 |
| | Frequenting places of interest | 🙂 | ☹️ | 🙂 | 🙂 | 🙂 | 🙂 | 🙂 |
| | Participation in events | 🙂 | ☹️ | 🙂 | 🙂 | 🙂 | 🙂 | 🙂 |
| RESTAURANT/ FOOD | Use of the water house | 🙂 | ☹️ | 🙂 | 🙂 | 🙂 | 🙂 | 🙂 |
| | SEND PHOTOS (doggy bag) | 🙂 | ☹️ | 🙂 | 🙂 | 🙂 | 🙂 | 🙂 |
| | QUIZ | 🙂 | 🙂 | 🙂 | 🙂 | 🙂 | 🙂 | 🙂 |
| TRANSVERSE TO ALL | Invite a friend | 🙂 | 🙂 | 🙂 | 🙂 | 🙂 | 🙂 | 🙂 |
| | Mission | 🙂 | 🙂 | 🙂 | 🙂 | 🙂 | 🙂 | 🙂 |

| LEGEND | ☹️ | 😐 | 🙂 | 🙂 |
|---|---|---|---|---|
| | NOT YET APPLIED | DURING PROJECT PHASE | STARTED | IN PLAN |

**Figure 13.** Monitoring of the impact areas introduced by the Bologna Charter.

The second level addressed the level of participation of the Metropolitan Member. The actual participation and the actions that the participants implemented the most, in the period between September and November 2021, were monitored. For example, in October 2021, there were 654 registered users, for a total of 357,579 Eco-points earned. From these, data emerges that the Metropolitan Members adhering to the Sustainability Charter were very active in participating in the good practices envisaged. With respect to sustainable sectors, the most implemented actions concern sustainable mobility, following waste recycling instructions and cultural ones.

The Sustainability Charter was funded by the Ministry of the Environment and Land and Sea Protection until December 2021 and will finish with the national competition in March 2022.

Actually, the Metropolitan City of Genoa, considering the high popularity and success so far, has decided to extend the initiative at least until the end of 2022.

Once this tool is fully operational, it would be interesting to further implement the app to investigate some more aspects. For instance, a catalogue could be added to the lottery mechanism that also includes prizes of low value (bus ticket, sponsored water bottle, etc.) to give everyone the chance to win a prize, while maintaining the draw of a high-value prize that attracts users.

Furthermore, it would be interesting to integrate the Eco-stop function via QR code. In this way, the various entities present in the metropolitan area that promote sustainable services with a dual function (service promoters and award providers) could be involved. For instance, a company that sells products at km0 could give the customer (EcoAttivo user) the possibility to frame a QR code after shopping or buying a product and, thanks to this function, obtain points, offer his own products at a discounted price, or for free as a reward.

The Sustainability Charter, developed in the paper, aspires to be a good practice: a scalable and adaptable solution to other administrative realities.

The purpose of the new proposed instrument falls within both the logic of achieving Agenda 2030 and the objectives set by the strategic urban plan of the city [86,90].

It is fundamental to build a sense of belonging to the metropolitan city area and increase attractiveness for businesses and investments, focusing on the enhancement of the productive, environmental, landscape and cultural excellence of the metropolitan-political area and on the improvement of quality of life, as an important competitive factor.

The project integrates the most effective diffusion techniques to involve users and spread the project on social networks. In particular, each user will see their daily performance, their points balance, their position in the national and municipal ranking, the badges earned through the different actions and the relative level of experience. The approach was structured to define incentive policies, rewards that come from the territory itself, valorising it and involving its inhabitants and its actors/stakeholders, which is fundamental to enabling the sustainability of the Sustainability Charter, and thus to achieve the goals that it intends to reach. Thanks to the involvement and raising of awareness, the population and the municipalities feel they are 'partners' and are equally responsible for a joint project that, in the long term, will hopefully produce a ripple effect that will attract increasing interest, capable of generating a virtuous circle towards sustainability.

**Author Contributions:** Conceptualization, Methodology, F.P., I.S., C.A., G.L. and P.G.; formal analysis, investigation writing—original draft, Preparation writing—review and editing, F.P. and I.S.; Validation, Supervision, Project Administration, Funding acquisition, C.A., G.L. and P.G. All authors have read and agreed to the published version of the manuscripts.

**Funding:** The project has been financed by the Ministry of the Environment on the basis of a public notice (decreto n. 344/2019) addressed to the Italian metropolitan cities in order to define strategic tools suitable to contribute to the realization/editing of the Metropolitan Agendas for sustainable development (activities as referred to in Article 34 of Legislative Decree no. 152/2006 and subsequent amendments.m.m.i.i.).

**Institutional Review Board Statement:** Not applicable.

**Informed Consent Statement:** Not applicable.

**Data Availability Statement:** Not applicable.

**Conflicts of Interest:** The authors declare no conflict of interest.

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
