# Peer review of "Application Studies for the Implementation of the Sustainability Charter in the Metropolitan City of Genoa"

_sustainability, doi:10.3390/su14084721_

Round 1
Reviewer 1 Report
The manuscript titled as “Sustainability Charter. The case of the Metropolitan City of Genoa” (Manuscript ID: sustainability-1634746) proposes a tool to apply sustainability in urban and regional planning. My main questions/concerns about the are as follows:
- The title should be reconsidered. It can be changed as “Sustainability Charter: The case of the Metropolitan City of Genoa”.
- The novel parts of the proposed tool should be more clearly stated.
- Figure 1 and 2 were not referred in the manuscript.
- The demonstration of Figure 5 should be improved. Putting the written parts carelessly on the circular arrows, made the figure looking bad.
- Discussion and Conclusions should be separated and the discussion section should be extended.
- The authors gave too much references to their previous studies and I think it is inappropriate.
Author Response
Thanks for your valuable suggestions. The answers are write in green below.
- The title should be reconsidered. It can be changed as “Sustainability Charter: The case of the Metropolitan City of Genoa”. The title has been changed
- The novel parts of the proposed tool should be more clearly stated. Sections 2 and 3 have been merged and improved according to your recommendations
- Figure 1 and 2 were not referred in the manuscript. Figure 1 and 2 has been referred in the manuscript
- The demonstration of Figure 5 should be improved. Putting the written parts carelessly on the circular arrows, made the figure looking bad. The figure was remade
- Discussion and Conclusions should be separated and the discussion section should be extended.
- The authors gave too much references to their previous studies and I think it is inappropriate. Some references have been deleted
In the paper, the main changes have been highlighted in yellow.
Reviewer 2 Report
Dear authors,
The article entitled “ Sustainability Charter. The case of the Metropolitan City of Genoa”, seeks to make an interrelation between the sustainability letter of the city of Genoa, Italy with the results of third-party applications that were evaluated by the authors. However, there is a serious problem of formatting and suitability of the article to scientific journals.
Following are the problems and adjustments pointed out for a better adaptation to what we can consider a publishable scientific article:
Title:
The title needs to be changed for the actual content of the article, a suggestion would be: "Application studies for the implementation of the sustainability charter in the metropolitan city of Genoa."
Summary:
Reformulate the abstract with the following information: objective of the study, brief exposition of what has already been studied and the gap found, the methodology, the main results, the discussion and the conclusions.
Key words:
They do not represent what the article actually is. Explore the use of keywords further, as up to five more suitable words can best represent the article.
Lines 32 and 33:
The introduction should give light to: global view of the subject addressed (contextualization), contemplate the relevance of the subject (justification), present why it was prepared (objective), present the research problem, mention previous works that address the topic in question and generically address what will be studied in the rest of the article (text structure).
Conclusions should come after discussion of the results and analyses.
Lines 118 and 119:
Explain in the introduction what the product was created and not just say that it was developed.
Item 2. Materials and Methods:
After the introduction, authors must create a specific item for the literature review.
The Materials and Methods chapter (item 2) can become a proposal or be merged with the Methodology chapter (item 3).
Line 140.
Very good. Location of sustainable development goals from the 2030 Agenda.
Line 142. Metropolitan Cities.
The authors must bring the concept of a metropolitan city, to characterize them well in difference to the other types of cities that exist in Italy.
Readers of the article will be from any part of the world and do not necessarily need to understand Italy's administrative and political structure. Adjust this point.
Lines 178 and 179.
Very good. The quintuple propeller when it turns leaves no one behind! Note that the Massachusetts Institute of Technology (MIT) uses the five-fold helix concept with investors being the fifth helix.
Figure 2. Lines 216 and 217.
At some point in the text, quote figure 2.
Lines 217 to 222.
Could the positive scoring system not resemble the score given by the Chinese to their population? The system can cause disparities. Example: those who do not have high scores live in conditions where they are often unable to get around or access a college.
Explain well the differentiation between citizen scoring programs with others existing in the world.
Item 3. Methodology.
Chapter 3 should explain the scientific method that underpinned the formulation of the proposal for the specific sustainability letter for Genoa.
Merge with chapter 2.
The text is not clear, the step by step needs to be better detailed. The presentation structure is not satisfactory.
Lines 232 to 235.
I don't understand the authors' statement. The map already exists and they are proposing a card, or do the authors propose the creation of a new map in cities of different dimensions (or with characteristics)? To explain better.
Lines 254 to 256.
I don't understand the authors' statement. The map already exists and they are proposing a card, or do the authors propose the creation of a new map in cities of different dimensions (or with characteristics)? To explain better.
Line 256. Phases.
For each phase, a specific sub-topic should be opened , explaining the phase, as well as its guidelines and expected results.
Lines 344 and 345. Monitoring.
sub-item must be opened after all other items (previous phases).
See Inter-American document Development Bank (IADB) that brings a methodology of traffic lights to monitor the evolution of sustainable cities.
Item 4. Application and Results.
Chapters four (4) and five (5) need to be reworked after a complete restructuring of the article, they are confusing.
Line 368.
The authors cannot explain whether it is a proposal or a product that has already been implemented, validated and is showing results. Readjust this point.
Lines 387 to 389.
The authors do not make it clear how they qualified the sample, the research instruments, the procedures and how other authors could reproduce the same experiment.
Did you actually build a product and validate a model or is it a framework in the testing phase?
Lines 422 to 424.
The authors make it clear that they did not produce the Letter of Genoa, but carried out an experiment with existing apps. The article needs to be redone and tell the truth, what was actually done by the authors and the results obtained.
Line 424 and 425. Method information should be taken to a methodology chapter. There are no results without proof of the methods used.
The article is biased and leads to a reading, shall we say, distracted and less judicious from the evaluators, to think that they created the Letter of Genoa.
The title itself suggests something completely different from what was carried out as research.
Results.
The results must be shown in a summary table, which allows any lay reader to understand what was researched, methodologically analyzed and what results were obtained. A simple format of inputs, processing and outputs.
Results analysis.
Discussions need to be based on the results obtained and not what is expected in a letter without direct implementation.
Future studies.
There are no future studies pointed out.
Data collect.
Authors must clearly demonstrate whether they have complied with the Helsinki protocol for collecting data from research participants.
Author Response
Thanks for your valuable suggestions. The answers are write in green below.
Title: The title needs to be changed for the actual content of the article, a suggestion would be: "Application studies for the implementation of the sustainability charter in the metropolitan city of Genoa." The title has been changed
Summary: Reformulate the abstract with the following information: objective of the study, brief exposition of what has already been studied and the gap found, the methodology, the main results, the discussion and the conclusions.The abstract has been better specified
Key words:They do not represent what the article actually is. Explore the use of keywords further, as up to five more suitable words can best represent the article. two other keywords have been entered
Lines 32 and 33:The introduction should give light to: global view of the subject addressed (contextualization), contemplate the relevance of the subject (justification), present why it was prepared (objective), present the research problem, mention previous works that address the topic in question and generically address what will be studied in the rest of the article (text structure).Conclusions should come after discussion of the results and analyses. The sentence has been moved to the conclusions
Lines 118 and 119:Explain in the introduction what the product was created and not just say that it was developed. Introduction has been better specified
Item 2. Materials and Methods:After the introduction, authors must create a specific item for the literature review.The Materials and Methods chapter (item 2) can become a proposal or be merged with the Methodology chapter (item 3). We have merged sections 2 and 3 and we have introduced a Subsection
Line 140.Very good. Location of sustainable development goals from the 2030 Agenda.
Line 142. Metropolitan Cities. The authors must bring the concept of a metropolitan city, to characterize them well in difference to the other types of cities that exist in Italy. Readers of the article will be from any part of the world and do not necessarily need to understand Italy's administrative and political structure. Adjust this point. We fixed this part but left the Italian case anyway
Lines 178 and 179. Very good. The quintuple propeller when it turns leaves no one behind! Note that the Massachusetts Institute of Technology (MIT) uses the five-fold helix concept with investors being the fifth helix. Ee have also added investors as possible actors. Thanks for the suggestion
Figure 2. Lines 216 and 217. At some point in the text, quote figure 2. Figure 1 and 2 has been referred in the manuscript
Lines 217 to 222.Could the positive scoring system not resemble the score given by the Chinese to their population? The system can cause disparities. Example: those who do not have high scores live in conditions where they are often unable to get around or access a college. . Very interesting .. add a reference to the Chinese system. Thank you
tem 3. Methodology.Chapter 3 should explain the scientific method that underpinned the formulation of the proposal for the specific sustainability letter for Genoa.Merge with chapter 2.The text is not clear, the step by step needs to be better detailed. The presentation structure is not satisfactory. we tried to improve the section following his instructions
Lines 232 to 235: I don't understand the authors' statement. The map already exists and they are proposing a card, or do the authors propose the creation of a new map in cities of different dimensions (or with characteristics)? To explain better. The sentence has been better specified
Line 256. Phases.:For each phase, a specific sub-topic should be opened , explaining the phase, as well as its guidelines and expected results.
Lines 344 and 345. Monitoring.sub-item must be opened after all other items (previous phases).See Inter-American document Development Bank (IADB) that brings a methodology of traffic lights to monitor the evolution of sustainable cities. we have proposed an approach for monitoring takes its cue from other projects that use the traffic light system (for a qualitative assessment) but adds further information on the level of implementation
Item 4. Application and Results.Chapters four (4) and five (5) need to be reworked after a complete restructuring of the article, they are confusing. the section has been modified in light of section 2
Line 368.:The authors cannot explain whether it is a proposal or a product that has already been implemented, validated and is showing results. Readjust this point. The sentence has been better specified
Lines 387 to 389.The authors do not make it clear how they qualified the sample, the research instruments, the procedures and how other authors could reproduce the same experiment. The sentence has been better specified
Did you actually build a product and validate a model or is it a framework in the testing phase? The test is concluded, but in this paragraph we have reported what we did before reaching the realization of the Sustainability Charter. Basically thanks to the Genoese case we have defined the guidelines. The guidelines in Chapter 2 introduce the procedures and how other authors could reproduce the same experiment
Lines 422 to 424.The authors make it clear that they did not produce the Letter of Genoa, but carried out an experiment with existing apps. The article needs to be redone and tell the truth, what was actually done by the authors and the results obtained. The sentence has been better specified
Line 424 and 425. Method information should be taken to a methodology chapter. There are no results without proof of the methods used.The article is biased and leads to a reading, shall we say, distracted and less judicious from the evaluators, to think that they created the Letter of Genoa.The title itself suggests something completely different from what was carried out as research. Some parts have been improved by moving them in the approach. Thanks for the suggestions in fact the theoretical methodological approach mixes with the application. But we assure you that we have created the Sustainability Charter!
Results.The results must be shown in a summary table, which allows any lay reader to understand what was researched, methodologically analyzed and what results were obtained. A simple format of inputs, processing and outputs.
Results analysis. Discussions need to be based on the results obtained and not what is expected in a letter without direct implementation. we have reported the numbers relating to the first months of application of the sustainability charter to the metropolitan area
Future studies.There are no future studies pointed out. Lines 597-610
Data collect. Authors must clearly demonstrate whether they have complied with the Helsinki protocol for collecting data from research participants. we have clarified this aspect better with the Journal.
In the paper, the main changes have been highlighted in yellow.
Reviewer 3 Report
The paper reports a search for application, with the purpose of defining the guidelines for a new Sustainability Charter created for a metropolitan-level city.
The main contribution is a review on the actions of communication and involvement of stakeholders (population, public and private actors) within the metropolitan area of Genoa trying to constitute the fundamental element of the new participatory tool of sustainability.
The Sustainability Charter was built around the actors involved, attracted the population to participate and achieve sustainability, changing together the wrong habits. Through presented project, sustainable entities had an opportunity to be more competitive and publicized.
The fundamental problem with this article is that it lacks some of the attributes of a scientific publication.
It contains neither scientific objectives nor hypotheses. Research methods are not specified. The literature used is current but contains a minimum number of scientific titles. In most cases, these are reports, documents, policies, agendas, plans, etc.
There are not mentioned sources under each figure.
For example FIG. 3 is not based on a strictly defined classification logic. E.g. in the Customer category are listed: student, resident, worker, tourist. What classification criteria did the authors follow?
The Service category states e.g. Mobility and Energy. From the point of view of scientific terminology (in the social sciences), this is by no means a type of service.
The manuscript is not entirely scientifically sound and the experimental design is not appropriate to test the hypothesis. The manuscript is quite clear, relevant for the field of sustainability, but is not presented as a well-structured scientific study. The article contains valuable results associated with the preparation of Sustainability Charter for the Metropolitan City of Genoa. I suggest that a manuscript may be more appropriate for publication in another journal where there is no strict emphasis on traditional scientific practices.
Author Response
Thanks for your valuable suggestions. The answers are write in green below.
We changed the title to make it more fitting with the paper. We have joined sections 2 and 3 where we report the scientific bases from which we started to structure the guidelines that have been applied to the case of the Metropolitan City of Genoa. We have tried to improve abstracts, introduction and the two sections of the approach and application.
In the paper, the main changes have been highlighted in yellow.
Round 2
Reviewer 2 Report
Dear authors, congratulations on the improvements in the article.
there are cases like figure three (3) that have empty columns and lots of text. Figure four (4) I consider it very polluted (a lot of text) and the phases of each stage should come in the body of the text and not in the figure.
For figure five (5) there is a lot of text on the cards that would need explanation and the figure would need to be redone to have an explanation for each piece of the proposed card.
In figure eight (8) the legend is out of line, as it should come with explanations below, in the body of the text.
Author Response
ALL THE REQUIRED CHANGES HAVE BEEN MADE.
Thanks again for the valuable reviews
Reviewer 3 Report
In Fig. 3, 5, 9, 10 and 13 the authors work with the term Service. Within this category, they use the terms Energy and Mobility among the service areas. As none of these cases are services, it is necessary to find another suitable term for defining these categories. E.g. in the case of mobility it can be a category of transport, in the case of energy eg energy conservation activities
Author Response

(The authors gave the same response as above.)
